# Bus Workers’ Experiences with and Perceptions of a Health Promotion Program: A Qualitative Study Using a Focus Group Discussion

**DOI:** 10.3390/ijerph17061992

**Published:** 2020-03-18

**Authors:** Jeehee Pyo, Mina Lee, Minsu Ock, Gwiok Park, Dongseok Yang, Jungsun Park, Yangho Kim

**Affiliations:** 1Department of Preventive Medicine, Ulsan University Hospital, University of Ulsan College of Medicine, Ulsan 44033, Korea; eesther0517@naver.com (J.P.); inyounsy@naver.com (M.L.); 2Department of Preventive Medicine, Asan Medical Institute of Convergence Science and Technology, Asan Medical Center, University of Ulsan College of Medicine, Seoul 44033, Korea; 3Ulsan Bukgu Contingent Workers Center, Ulsan 44033, Korea; tgbear@naver.com; 4Department of Physical Medicine and Rehabilitation, Ulsan University Hospital, University of Ulsan College of Medicine, Ulsan 44033, Korea; fnew11@gmail.com; 5Department of Occupational Health, Catholic University of Daegu, Gyeongsan 38481, Korea; jsunpark@chol.com; 6Department of Occupational and Environmental Medicine, Ulsan University Hospital, University of Ulsan College of Medicine, Ulsan 44033, Korea; yanghokm@uuh.ulsan.kr

**Keywords:** health promotion program, bus workers, focus group discussion, vulnerable workers

## Abstract

We conducted a focus group discussion with bus workers to evaluate their experiences and perceptions, as well as their reasons for participating in a program for vulnerable workers. The study also sought to identify the strengths and weaknesses of the program. A total of nine bus workers participated in the focus group discussion. The focus group discussion was conducted based on a semi-structured guide, which was developed from discussions among the researchers and a review of major preliminary studies. The verbatim transcriptions were analyzed using content analysis. The sole existence of a health promotion program for vulnerable workers in Ulsan was sufficient to attract participants who had not been involved in other health management programs. Participants reported that participation in the program mitigated some of their musculoskeletal symptoms, which are often faced by bus workers. The findings of this study may contribute to the expansion of the health promotion program targeting workers in vulnerable working environments in Ulsan and other places. In order to increase users’ satisfaction and the sustainability of the program, it is necessary to develop a strategy to increase the accessibility of the program and enhance the self-efficacy of participants.

## 1. Introduction

A vulnerable worker refers to a laborer whose well-being is at risk as a result of being involved in the labor market [1]. Vulnerable workers include day and temporary laborers, simple laborers, elderly laborers, and female laborers—all of whom are in precarious employment arrangements. They are more likely to face health problems [2], are exposed to the danger of industrial accidents, and are often excluded from the preventive or protective measures against industrial accidents [3]. This could be primarily attributed to an unstable and meager labor environment of the vulnerable workers [4]. Furthermore, reports of the health-related status of vulnerable workers highlight a higher level of prominent unhealthy behaviors such as smoking [5] and drinking [6] among vulnerable workers compared to those in other labor conditions. Therefore, it is necessary to stabilize the labor market, improve the working conditions of the vulnerable workers, and monitor and prevent their health issues. It is also essential to develop and implement interventions that can promote healthy behaviors among vulnerable workers. 

Transportation workers, including bus workers, are exposed to risk factors such as vibration, noise, and hazardous chemicals. Moreover, they are often required to work for long hours without guaranteed breaks in a cramped and small working space [7]. They follow a shift system during dawn or late hours. In particular, bus workers work in a specific environment that can lead to more difficulties, such as stress from extreme dispatch interval times, the weight of potential accidents, and the responsibility of passengers’ safety [8]. As a result, numerous passenger traffic and freight transport businesses are faced with disasters such as occupational off-site traffic accidents, falls, crashes, and cardiovascular diseases [9]. In other words, transportation workers such as bus workers need to be regarded as vulnerable workers with a relatively high risk of industrial accidents, which warrant the need for urgent healthcare management interventions. 

The Ulsan Metropolitan City Bukgu Contingent Workers Center has been preparing to embark on a project to create an environment in the Ulsan area where small-scale workplaces with less than 50 employees can labor in good health, by bridging resources such as health, medical care, and safety preparations. It is part of the activities for the revitalization of the regional labor, management, citizen, and government cooperation project since 2014 [10]. Notably, the project has been selected as the “Improvement of Employment Quality and Elimination of Welfare Blind Spot” among Ulsan-specific job innovation projects since 2018. It is being expanded and operated as a worker health promotion project for Ulsan-based vulnerable workers. A consensus on endeavoring to develop a program beyond the typical support healthcare programs for workers in small establishments with vulnerable working environments has enabled this new project. It is expected to advance the labor quality and stabilize and sustain the employment status. The health promotion program for vulnerable workers established in 2018 constitutes three programs: leaders in the health training program for apartment cleaners; participatory work conditions improvement project for school food service workers; a health promotion program for industrial manufacturing and transportation (bus and cargo) workers.

The present study focused on the third program—health promotion program for bus workers. In 2018, the program offered stretching training for musculoskeletal disorder prevention, exercise prescription, and counseling based on the results of a general medical examination for about three months. Rehabilitation specialists provided stretching training and internal medicine care, and other specialists referred to the medical examination results. The specialists regularly visited the main bus garages in Banguhjin, Nongso, and Yuli, Ulsan to provide direct services. In addition, the program provided self-exercising tools to encourage self-stretching, brochures containing healthcare guidelines for promotional and educational purposes, and posters with stretching instructions, which were installed in the employee area. Moreover, a stretching promotional video was created and distributed to bus drivers.

For this study, a focus group discussion with bus workers was conducted to evaluate their perceptions of, and experiences with, the program, motivation for participation, and the strengths and limitations of the program. The findings of the present study are expected to provide a better understanding of the program, which can help to enhance it in the future, as well as understand its implications for other similar programs in Ulsan.

## 2. Materials and Methods

In this qualitative study, a focus group discussion was conducted with bus workers to evaluate their perceptions and experiences of the program, in order to explore strategies to further improve the program. The consolidated criteria for reporting qualitative research were applied to describe the qualitative research process using a focus group discussion [11].

### 2.1. Research Team

A total of three analytical and four auditing researchers constituted the research team. The analysis team comprised of one preventive medicine specialist, one researcher, and one clinical nurse with 11 years of experience. The audit team included one occupational environmental medicine specialist, one rehabilitation medicine specialist, and one small-scale business executive. Two of the researchers in the analysis team have extensive experience in qualitative research; they are experienced in reporting qualitative studies in research papers and theses, have attended seminars and conferences, and have conducted lectures related to the method.

### 2.2. Research Participants

Nine bus workers attended a focus group discussion. The small-scale business executive contacted them individually to explain the purpose and content of the study. Then, those who agreed to participate were selected. Response bias, which refers to the tendency of respondents to respond according to researchers’ intentions, is likely to be low, given the research topic and purpose.

### 2.3. Progress of Focus Group Discussion

The focus group discussion was conducted according to a semi-structured guide developed from discussions among the researchers and a review of major preliminary studies. The guideline comprised the following four aspects: bus workers’ awareness of the health promotion program; motivation for participation in the program; perception of the contents of the program; opinions and suggestions for the improvement of the program. Prior to the discussion, participants were fully informed about the purpose of the study, and therefore, verbal consent was obtained. One researcher conducted the discussion, which took about two hours. The discussion was recorded and transcribed for analysis. 

### 2.4. Analysis

Content analysis was performed to analyze the data [12]. The transcribed contents of the focus group discussion and the researcher’s notes were used for the analysis. Content analysis involves extracting, reinterpreting, and inferring significant implicit meanings embedded in original data by using established theories or perspectives. It is categorized as conventional content analysis, direct content analysis, and summative content analysis according to the approach. In this study, directive content analysis was adopted; it refers to the categories derived from existing theories or the results from the analysis. The guide developed for the focus group discussion was used in the analysis.

A detailed analysis was performed, and concepts were derived by a researcher through repeated reading of the transcribed contents and notes of the focus group discussion. The derived concepts were then reviewed by another researcher, and conflicting opinions among the two researchers were resolved through sufficient discussions. Another researcher examined the concepts derived by the two researchers, who participated in the primary analysis and categorized similar concepts from among the ones resulting from agreements. All researchers then examined the results of the analysis, focusing on the framework of the entire category. The participants did not review the results of the analysis, and four supervisors finalized the assessment of the analysis results.

## 3. Findings

### 3.1. Demographic Characteristics

The characteristics of the study participants are shown in Table 1. There was one female and eight male participants. In terms of age, three participants were in the 40s, five in the 50s, and one in the 60s. 

### 3.2. Analysis Results

A total of 100 concepts were derived and categorized into the four themes: obstacles to the healthcare of bus workers; the first positive impression of the health promotion program; satisfaction and changes after participating in the health promotion program; improvement of the health promotion program. Table 2 summarizes the results of the analysis. The following sections describe the core contents according to each category and subcategory.

### 3.3. Obstacles to the Healthcare of Bus Workers

In general, bus labor is characterized by working in the same position for a long period of time in a limited space (a bus) [7]. This working environment can lead to the onset of not only cardiovascular diseases but also musculoskeletal pain in workers [9]. The city bus companies, in which the participants work, follow a two-shift system and have short dispatch intervals (participant 1, 2, 6, 8). These working conditions may be related to health problems such as poor physical fitness, digestive disorder symptoms, sleep apnea, and poor sleep quality (participant 1, 2, 4, 5, 6, 8, 9).

#### 3.3.1. Irregular Meal Schedule Due to Lack of Time 

The participants were not able to consume meals regularly due to the two-shift working system and the short dispatch intervals. They had two meals per day and failed to establish a regular meal schedule (participant 1, 2, 6, 8). Moreover, it had become normal for the participants to rush meals due to the tight dispatching interval time. This resulted in chronic indigestion (participant 2) and abdominal obesity (participant 6, 8) among the participants, which eventually deteriorated their health.


*Participant 9: As you know, we have a designated amount of time to eat. There should be at least 20 to 30 min to eat, but we only get ten minutes sometimes.*



*Participant 6: I eat in two minutes.*



*Participant 9: I pour water into my rice bowl—that is the environment (where I work at).*



*Participant 8: ... We do not have time to eat. We just scoop out the rice out and eat it. It’s like when we ate rice when cannonballs were falling during the Korean War.*



*Participant 8: In the morning, when I say (to my wife) that I working the afternoon shift, I get a meal at 7 or else 10 or 11 AM.... Then I return from the shift, and I eat around 3 or 3:30 PM… then, I eat at 7 PM.… and I eat again at 11 PM.…I need to be on a diet. I mean, my belly is sticking out. If you go through this (schedule) for three months, you will get this stomach too; it is for real. I have no other choice because it is all in the contract.*


#### 3.3.2. Mental Difficulties due to the Shift-Work System 

The majority of the participants complained of sleep disturbances due to the shift-work system (participant 4, 5, 6, 8, 9). It was a major predicament, as their daily lives could only function by receiving psychiatric counseling or taking medication (participant 9). In addition to the work schedule, having a direct association with passengers, which was a job characteristic, was also a factor that impaired the mental health of the participants. They stated that they experienced significant difficulties due to inappropriate reproach and verbal abuse in an environment directly disclosed to the passengers, which made them want to quit their job, ultimately (participant 1, 7). 


*Participant 6: I have a sleep disorder.*



*Participant 9: One time I could not sleep for three days because I did not take the pill. I take zolpidem or something… I know I should not, but I need to work. I have to sleep in some way - I called in sick too… I do not want to take it either.*



*Participant 5: I see myself as a disabled person. I have been suffering from insomnia. It has been a few years now. The more you take it, the more damaged you are.*



*Participant 1: A friend my age, who has similar characteristics as me, could not ignore and let go of the wrongdoings of passengers. I heard he received psychiatric treatment and eventually quit. What I am saying is that, these things made me extremely stressed; bus driving is not only a physically strenuous labor but also a mentally stressful job…*


#### 3.3.3. Lack of Awareness of Bus Workers’ Healthcare

The participants continued to maintain a naïve attitude towards health issues, despite their declining physical and mental health. Most participants acknowledged the necessity of healthcare but continued being negligent of their health due to lack of time (All participants). Furthermore, they were not aware of the accurate figures for their blood pressure and blood sugar levels, to prevent cardiovascular diseases, which are likely to be caused by their work environment (e.g., shift-work, sitting posture, conflict with passengers) (participant 1, 6, 7, 8, 9). Moreover, they recognized the discrepancy in their knowledge on basic self-healthcare, although they made very few attempts to ameliorate it. They believed that perceptions of healthcare vary with age and willingness (participant 1, 2, 3, 6, 9). That is, the higher the age, the more interested in health status one is. Engaging in health-related behaviors depends on the will of a person. Therefore, they anticipated that positive reformation of a work environment would not have any effect on their health (participant 7).


*Participant 8: It is no fun when you do it alone.*



*Participant 4: I use the gym equipment on the riverside when I have the time, but to be honest, I do not really work out at home.*



*Participant 8: I must do it when I have time, but I just do not. Twenty-four hours is not enough.*



*Participant 5: I do not have the time for it.*



*Participant 3: Younger people in their 20s and 30s are not interested at all. They are reluctant and feel awkward to get involved in a program like this.*



*Participant 7: I personally think that the will of a person is the most important regarding matters of health, whether it is about using a handbook or an app (smartphone application) on health. I think that no more than 30% of workers will use it.*


### 3.4. The Positive First Impression of the Health Promotion Program

The participants did not have any previous experience with other health promotion programs, despite their extensive career (All participants). Some participants (participant 6, 9) engaged in exercise as a hobby and managed their health without considering the health issues arising from work. Under these circumstances, the present health promotion program was sufficient to attract the attention and involvement of the participants in the program (participant 3, 4, 5, 6, 8, 9).

#### 3.4.1. No Experience Participating In Prior Health Promotion Programs

All participants recognized that their physical and mental health was deteriorating due to the detrimental working environment. Some of the participants managed their health through fitness and soccer during their days off (participant 6, 9). These activities, however, did not contribute to the improvement of the musculoskeletal symptoms, a constant inconvenience sustained by their work. None of the participants had any prior experience of participating in health promotion programs, including programs held by public health centers or employee health support centers.


*Moderator: Have you ever tried a similar program like this? Have you tried it before?*



*All participants: No, I have not.*



*Moderator: You are aware of a public health center or an employee health support center, I assume? Have you ever participated in any similar programs there?*



*All participants: No, I have not.*



*Participant 9: This is the first time…the first time since 1991, when I started my job.*


#### 3.4.2. Motivation to Participate in the Health Promotion Program

There were various reasons for participants to join the health promotion program. Most of them took an interest in the program after seeing the promotional banner or promotional articles on social networks (participant 3, 4, 5, 7, 8). In addition, some voluntarily joined in hopes of relieving their physical pain (participant 6, 9).


*Moderator: How did the rest of you hear about the program?*



*Participant 6: I saw the banner.*



*Participant 5: … It could be spotted at the garage.*



*Participant 6: I received a text message.*



*Participant 3: There was an announcement in my company, and for some other companies, a union leader posted the announcement on the union social network.*



*Participant 9: I thought I should give it a try. When I saw it, nobody told me about it, but I wanted to give it a shot when I spotted it.*



*Participant 6: I thought it could be helpful because my neck and shoulder ache…*


### 3.5. Satisfaction and Changes after Participating in the Health Promotion Program

Participation in the program was an opportunity to reassess the health problems that the participants had been neglecting (participant 3, 5). Participants reported experiencing positive changes and significant satisfaction (participant 1, 2, 5, 7, 9). They were able to gradually relax the persistent pain through quick stretching and understood their current health status through various medical examinations (participant 1, 2, 7). The lack of publicity for the program, however, generated a low participation rate. This phenomenon was regarded unsatisfactory, which led to the participants encouraging their peers to join (participant 1, 6, 9).

#### 3.5.1. Positive Changes as a Result of Participating in the Health Promotion Program

The numbers of involvement in the health promotion program varied by the participants from one to eight-time involvement, but they experienced a positive transformation from the stretching skills they acquired. The most significant change was that the stretching skills enabled them to take health-related actions in daily life (participant 1, 2, 5, 7, 9). In the long-term, it is expected to improve their healthcare behavior and health status. They also recognized a shift in their perception of the working condition after participating in the program (participant 1, 9). Before the program, they considered the working environment and the short dispatch intervals to be the significant factors, which forced them to feel helpless about healthcare. However, they learned to utilize the workspace to stretch efficiently for a short amount of time. From this understanding, they established a routine, which comprised stretching, and observed improvement in their physical discomfort. This encouraged them to participate in other available health promotion programs (participant 3, 5, 7).


*Moderator: Is it better after the stretching? Can you feel it?*



*Participant 5: I had some shoulder pain, and there were two physical therapists; two of them gave me shoulder massages, and I got them about three times. Then, I did it at home, and it was quite pleasant.*



*Participant 9: (I have a) frozen shoulder. I have terrible shoulder pain, and I felt great relief doing it.*



*Participant 2: Yes, it felt untangled.*



*Participant 9: They also gave tools. If you do what you learn from them, you could actually pull it through. Because my body is in pain…*



*Participant 1: I do it whenever I start the engine.*



*Participant 9: Yes, it helped a lot.*


#### 3.5.2. Satisfaction Factors of the Health Promotion Program

The program primarily focused on the prevention of musculoskeletal disease through stretching and exercise and included examinations of the body fat, body mass, osteoporosis, blood pressure, blood sugar, and cholesterol. Such a comprehensive healthcare program resulted in high satisfaction among the participants (All participants). Furthermore, considering the lack of free time, they were content with the availability of health counseling and individualized posture correction for painful areas from professionals, in their workplace (bus garages) (participant 1, 3, 7). Moreover, they were satisfied with the strategy to associate the stretching tools, compensation of the program involvement, engaging in a daily healthcare routine, and the companionship of colleagues throughout the engagement (participant 8).


*Moderator: You have had various experiences. You underwent blood drawing and stretching exercises, but which one was the best of all?*



*Participant 2: It was the body fat analysis.*



*Participant 1: I went for that (body fat analysis) to see the body fat percentage and to be assessed. I wanted to know my body measurements, and therefore, I was recommended to colleague.*



*Participant 3: Things you could conveniently do, like stretching, by seeing a doctor. You feel like a patient when you visit a hospital, but you could go there (garage) without any pressure. I can casually go there, and at the same time, I can receive professional diagnosis and treatment.*



*Participant 1: I had easy access. For so-called-hospitals, you have to visit deliberately and that is inconvenient. However, there (garage), you could just do it for just being there.*



*Participant 9: I tried the gift (stretching tool) at home, and it was really good. It stretched my back, and I experienced immense relief.*



*Participant 7: I used to have shoulder pain when exercising. It was damaged, so I stopped it, but the pain did not subside. Therefore, I went to an orthopedic, but the doctor told me that nothing was wrong with me. So I just let it slip. … I clearly did not know how to treat it…during the stretching activity, I was informed about an injured area that I was not aware of. It was not the most impactful exercise, but I wish they taught me muscle or ligament stretching movements…because lately, it has made me feel much better.*


#### 3.5.3. Limitations of the Health Promotion Program

The participants expressed great satisfaction with the program but noted the limitations as well. One of the most mentioned points was the lack of program publicity (participant 1, 6, 9). Participants who reflected on their health and experienced improvements through the informative sessions despite the short-term period felt sorry for their colleagues, who could not get involved due to the lack of publicity (participant 1, 6, 9). Moreover, some of them worried that the unmodifiable working environment would eventually worsen the symptoms regardless of the stretching skills and felt unsatisfied with an insufficient health consultation (participant 3, 8).


*Participant 9: I think with good PR (public relations); I mean many workers did not do it simply because they did not hear about it. But if you were to recommend it to your co-workers by saying “It is really nice when you actually try it out” and “Just try it! It is truly helpful!” then they might become willing to participate. But the PR was almost invisible, which made the workers hesitate. You need proper promotion for this.*



*Participant 3: It is a good idea to hop on a car and stretch. However, the unpleasant part, rather than feeling better from the exercise is that… if you get an old car, you would feel the symptoms worsen quickly, rather than improving.*



*Participant 8: I benefitted from the exercising or other activities, but I did not benefit from, for example, finding out how good or bad my health was.*


#### 3.5.4. Recommendation of the Health Promotion Program to Colleagues

The great fulfillment of the program led to participants recommending the program to other colleagues (participant 1, 5, 6). Participants shared their positive involvement in the program with their peers who could not join because of insufficient advertising. They believed that those who have experienced the program should enthusiastically inspire others to participate (participant 1, 6).


*Moderator: What would be a good PR strategy for this (addressing the low participation rate)?*



*Participant 6: We should actively promote it.*



*Participant 1: People who have participated in this program can spread the word, but people who have not cannot; if more people participated in the program, then the word would have traveled faster.*



*Moderator: Would you consider recommending this program to others?*



*Participant 1: Yes. If your body gets better, of course, you will recommend it.*



*Participant 5: I encouraged a lot of them by saying things like “It is nicer than I expected.”*


### 3.6. Improvement of the Health Promotion Program

Participants mentioned several aspects of the program that could be enhanced. The current program manages only physical health. They emphasized the importance of sleep disorders and mental predicament as a result of their work environment and the inclusion of a mental health consultation session in the program (participant 1, 6, 7, 9). Moreover, they asserted that the continuity of the program would amend the core of the problem lied in health (participant 2, 6, 8, 9). Furthermore, the persistent obstacles as a result of the two-shift work system and short dispatch intervals were other compelling reasons for the low participation rate (participant 1, 3, 5, 6, 8, 9). They highlighted the necessity of adopting a unique shift system for a smooth implementation of the program (participant 1, 2, 3). They also noted that maintaining a health handbook or distributing health-related posters could serve as adequate substitutes in case of participation failure (participant 2, 3, 5, 6, 8, 9). They also stated that not only should the program be expanded to other regions, but also small establishments with less than five employees should be prioritized for the program (participant 3, 5, 8). 

#### 3.6.1. Expectation of the Inclusion of Mental as well as Physical Healthcare in the Program

Mental health problems were an imperative factor deteriorating a healthy routine. The physical pain and mental difficulties could remain unaddressed and eventually threaten the responsibilities of the worker (participant 6, 9). Chronic pain and fatigue accumulation from working in such an environment can decrease the concentration level and increase sleep disturbances (participant 4, 5, 6, 8, 9). In addition, unfair treatment and criticism from passengers could trigger mental stress, which could hinder bus operation (participant 1, 7). They therefore strongly argued that mental difficulties should be managed to avoid operational complications (participant 1, 6, 7, 9).


*Participant 1: We (workers) talk a lot about grievances when we take a break in the employees’ room. This has something to do with psychological wellbeing as well … it might be quite helpful to get some counseling or… Because I am extremely stressed out, bus driving is not only physically taxing but also psychologically stressful. Therefore, psychological counseling is needed.*



*Participant 9: It might be because of my age, but I have not been able to sleep since last March; sleeping does not work on me anymore. So I consulted a few psychiatrists… strangely, if I wake up in the middle of the night, I cannot go back to sleep. Moreover, I have been suffering for one year and seven months. I am seeing a counselor too…*



*…*



*There are more ways than I thought. They just keep their mouth shut and keep it to themselves. Rather than just telling us to lower the accident rate, I wish we could have some peace of mind. Loss of sleep is often directly associated with accidents, as you know.*


#### 3.6.2. Continuous Health Management Support from Professionals

Many participants pointed out the inevitable need for a regular program rather than a one-time event (participant 2, 3, 6, 8, 9). A sudden healthcare event is not practical under circumstances where it is challenging to transform the working condition (participant 8). They also claimed that there is a need for permanence to understand their health status and maintain a proper stretching posture (participant 2, 3, 6, 8, 9).


*Participant 8: It is an only a one-time thing if you do it for a week and you rest a bit. … I want the current program to be gradually developed and continued for a longer period of time.*



*Participant 6: I wish it became a long-term program.*



*Participant 3: I think it would be appropriate to make things like this last for a while…*


#### 3.6.3. Carrying out the Program in Consideration of the Working Conditions

Participants’ irregular working hours and short dispatch interval times hindered program engagement (participant 1, 3, 5, 6, 8, 9). Participants whose pain improved through the program attempted to get more involved in the program (participant 1, 6, 9). However, inconsistent working and program hours led them to neglecting their health again (participant 1, 9). They inquired about operating hours at various times and wished to be informed of the program’s location and schedule through social networks (participant 1, 3, 5, 6, 9). 

#### 3.6.4. Need for Including Training for Improving Self-Healthcare Skills in the Program

They presented a positive response to a healthcare handbook as a way to self-manage health (participant 2, 3, 5, 6, 8, 9). Considering the relatively high age range of the bus workers, they expected that the handbook would suit them better than an unfamiliar smartphone application and that it would provide them an opportunity to manage their health without being involved in the program (participant 2, 5, 6, 8). Furthermore, some considered that the posters at the workplace would naturally encourage stretching activities, which would enhance the effectiveness of the program (participant 8).


*Moderator: What do you think about being provided with a health handbook so that one can keep a record of their health?*



*Participant 9: It would be nice if you could give it out.*



*Participant 6: It is good to keep a track; I will know when and what I did, and I can reflect on it too.*



*Participant 2: I will have the “Ah ha! I got it!” moments.*



*…*



*Moderator: What about a smartphone application? Do you generally use it well?*



*Participant 2: It is okay, but I would prefer a handbook.*



*Participant 5: Me too.*



*Participant 6: People like us do not really know…youngsters do though…*



*Participant 8: Remember when there was a national gymnastic exercise or some sort of an event with instructions? I think it would be more effective if you stick a poster on a wall and follow it regularly whenever you have the time.*



*Moderator: Are you suggesting making a poster about it?*



*Participant 8: Indeed. Even I would follow it naturally if it was in the office.*


#### 3.6.5. Expansion of the Health Promotion Program

The participants felt positive about the expansion of the program. They suggested prioritizing small-scale businesses with less than five employees as the companies often do not provide proper healthcare for financial reasons (participant 3, 5, 8). Moreover, they recommended the expansion of the program to other regions (participant 2, 8).


*Participant 5: Where should it start… There are tons of small businesses. Regular small businesses go through a health checkup process once a year. But smaller ones hardly do it. There are just thousands of them out there.*



*Participant 8: You have no idea how many companies with three to four people are there… Never in their wildest dreams would they go to a hospital.*



*Participant 5: No, never.*



*Participant 2: I wish other regions will get a chance to experience it too… The good part of the program is that stretching is essential, but it is not so easy to do it as you picture it in your head. If you squeeze it in every second, I think it would be nice to see the program in every region.*


## 4. Discussion

A focus group discussion of bus workers was conducted to examine their experiences and perceptions of the health promotion program, in order to gain a better understanding of the program and improve it. The health problems of transportation workers was highlighted for select periods. Precisely, it was reported that the likelihood of experiencing physical disorders such as cardiovascular diseases [13,14], musculoskeletal disorder [15], noise-induced hearing loss [16], stroke [17], as well as psychiatric disorders such as depression and anxiety [18], and occupational stress [19] are higher in such workers, relatively. There is, however, still a shortage of health promotion programs in South Korea for bus workers. Although some studies have investigated the effectiveness of several healthcare programs in small establishments [20,21,22,23], qualitative studies examining the overall experiences and perceptions of participants in health promotion programs are limited. The present study is significant because it highlights not only bus workers’ experiences with the health promotion program, which also is the first health promotion program for bus workers in South Korea, but also the overall effectiveness of the program.

Four themes were obtained in this study: obstacles to the healthcare of bus workers; the first positive impression of the health promotion program; satisfaction and changes after participating in the health promotion program; and improvement of the health promotion program. First, based on previous studies, it was confirmed that bus workers are at risk for several health problems. In particular, many of these health problems are caused by the working environment of bus workers. Participants complained of severe digestive problems and sleep disorder due to the tight working hours and the two-shift working system. They also mentioned mental stress caused by conflicts with passengers. They were aware of their healthcare needs to some extent, but lack of time was one of the reasons for neglecting their health. As the health of the bus workers is directly related to passenger safety, it is necessary to pay more attention to improving the working environment, which can fundamentally solve the health problems of bus workers [24]. It is also necessary to develop effective interventions that allow bus workers to recognize the need for care [25]. The worker health promotion program for the Ulsan-specific worker environment vulnerable group appealed to participants who had not been involved in other healthcare programs and were sufficiently aware of their healthcare needs. There is, indeed, a lack of publicity and participation still, but we obtained a glimpse of the needs of the bus workers’ health promotion program and the expectations on the contents of the program. On the other hand, it is essential to examine whether the programs that are already being implemented at public health centers or workers’ health support centers are satisfying the needs of current bus workers or not. Bus workers with time constraints might prefer programs that are conducted at the workplace itself, where the healthcare activities are also carried out, over programs that require visiting the public health centers or workers’ health support centers. Therefore, it is anticipated that the participation of bus workers and other workers in the program will be improved by expanding and managing the program to the workplace and implementing it continuously [26].

The present health promotion program was reported to mitigate some of the health problems faced by bus workers. It was found that stretching exercises for bus workers, who are required to work in minimal space, could prevent musculoskeletal disorders [27,28], and they stated that they experienced the positive effects of stretching, which increased their satisfaction with the program. Furthermore, the rehabilitation specialists and physiotherapists provided posture correction education to reduce pain in specific areas of the body, which also helped participants to feel more relaxed. Many healthcare educational programs are currently being held in the form of collective education, and it is likely that one-on-one educational sessions can improve participants’ satisfaction as well as the effectiveness of the educational program [29]. Furthermore, the stretching activities not only alleviated musculoskeletal pain but also enabled bus workers who lack time to efficiently manage their healthcare activities even in a short period of time. In other words, it increased bus workers’ self-efficacy for healthcare as a result of the positive effects of the stretching activities. Thus, health promotion programs in the future should be designed to provide immediate positive effects. 

The suggestions for the improvement of the health promotion program could be categorized as content- and operation-related improvements. Suggestions for content-related improvements included adding more health promotion activities to address various other health problems. Notably, it was found that there was a high demand for activities for sleep disorder and stress management. Stress management of bus workers is essential in situations where there is always a risk of traffic accidents, and the possibility of conflict with passengers and other workers [30]. As bus driving has a direct relation to the safety of the passengers, there is a need to develop measures to manage the mental health of bus workers. Future health promotion programs for workers in Ulsan-specific vulnerable working environment should consider the following in expansion: including factors that prioritize the management of mental health and establishing a system to prevent and manage the mental health of transportation workers through connecting them with municipals and mental health promotion centers.

Regarding suggestions for operation-related improvement, participants stated that the core health problems could be addressed if the program was implemented continuously rather than as a one-time event. The continuous implementation of the program is also crucial for building rapport with the participants and also for expanding the range of participants. Moreover, considering participants’ work environment, it is necessary to improve the program’s accessibility by, for example, providing more program hours and introducing a reminder system via social network. Unfortunately, due to time and budget constraints, it might be challenging to considerably increase the accessibility of the program. Thus, developing a strategy to increase the accessibility of the program in order to increase participation and satisfaction with the program is required; it is necessary to develop strategies to improve bus workers’ the self-efficacy for healthcare. For example, it is crucial to consider how to incorporate various activities into the program, such as providing a healthcare handbook that allows participants to track their healthcare behaviors, or reminding participants, directly and indirectly, about the necessity and methods of stretching before work.

A limitation of this study is that it could not quantify the effects of the health promotion program. Based on the findings of this qualitative study, it is expected that this health promotion program will help to mitigate musculoskeletal symptoms of bus workers, prevent symptoms from worsening, and lead to a higher participant satisfaction. Nevertheless, an understanding of the quantitative effects and satisfaction is necessary for the expansion of the program. Therefore, in the future, it is essential to evaluate the effectiveness of the health promotion program based on the contents and measurements of the questionnaire before and after participation in the program. Another limitation is that the study did not confirm the opinions of bus workers who did not participate in the program. In order to enhance the accessibility of the program, future studies should include bus workers who have not participated in the program previously. This study had only one female participant, which is also considered to be a limitation. Although the experiences and awareness of female bus workers in a male-dominated workplace are expected to vary, it is difficult to analyze and interpret different results given the small sample size in this study. In future studies, it is necessary to examine the perception of healthcare of female bus workers.

## 5. Conclusions

This qualitative study provides an understanding of the extent of bus workers’ health problems and their healthcare status as well as the effectiveness of the program, and provides information on how it can be improved. The study findings are expected to explain the importance of workers’ health problems and to expand health promotion programs targeting workers in vulnerable working environments in other regions.

## Figures and Tables

**Table 1 ijerph-17-01992-t001:** Characteristics of the participants.

N	Gender	Age Group (Years)	Work Place
1	Male	50s	‘A’ bus company
2	Female	40s	‘A’ bus company
3	Male	50s	‘B’ bus company
4	Male	50s	‘C’ bus company
5	Male	50s	‘A’ bus company
6	Male	40s	‘A’ bus company
7	Male	40s	‘D’ bus company
8	Male	50s	‘A’ bus company
9	Male	60s	‘A’ bus company

**Table 2 ijerph-17-01992-t002:** Categorization results.

Main Category	Subcategory
1. Obstacles to the healthcare of bus workers	1-1. Irregular meal schedule due to lack of time
	1-2. Mental difficulties due to the shift-work system
	1-3. Lack of awareness of bus workers’ healthcare
2. The first positive impression of the health promotion program	2-1. No prior experience of participating in health promotion programs
	2-2. Motivation to participate in the health promotion program
3. Satisfaction and changes after participating in the health promotion program	3-1. Positive changes as a result of participating in the health promotion program
	3-2. Satisfactory aspects of the health promotion program
	3-3. Limitations of the health promotion program
	3-4. Recommendation of the health promotion program to colleagues
4. Improvement of the health promotion program	4-1. Expectation of the inclusion of mental as well as physical healthcare in the program
	4-2. Continuous health management support from professionals
	4-3. Carrying out the program in consideration of the working conditions
	4-4. Need for including training for improving self-healthcare skills in the program
	4-5. Expansion of the health promotion program

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
