# Peer review of "Bus Workers’ Experiences with and Perceptions of a Health Promotion Program: A Qualitative Study Using a Focus Group Discussion"

_ijerph, 2020, doi:10.3390/ijerph17061992_

Round 1

Reviewer 1 Report

First of all, thank you for giving me the opportunity to review this interesting manuscript on the perceptions of workers in vulnerable situations.

The study follows an appropriate methodology to resolve the objectives set, and clearly sets out the results obtained. It is a relevant and well-conducted study. However, there are some minor aspects that should be taken into account before its publication.

1) The section on results exposes too many subjective considerations written by the authors themselves that should be resolved in the discussion section instead of being presented together with the direct opinions of the participants. It is recommended to review this aspect and reduce the subjective comments of the authors in this section.

Author Response

We have attached a review comments response file.

Reviewer 2 Report

I commend the authors for undertaking a very interesting focus group discussion with bus drivers on their perceptions and experiences of a health program. Even more impressive, is the fact that this is the first health promotion program for bus workers in South Korea.

Please consider the specific comments and suggestions below:

Introduction: Lines 35-46 I would request that the authors consider removing this section of the introduction, because the focus of the paper is on the vulnerable working environment, rather than the vulnerable worker. If indeed, the bus workers are vulnerable, then please state which workers had an underlying health condition in the 'characteristics of participants' table.

Please also consider the use of vulnerable workers in the abstract.

Line 81: … to evaluate their perceptions...

Material and methods: 

Line 88: their perceptions and experiences of the program, ...

Line 102: this statement refers to response bias, not reflexivity.

Line 111: transcribed verbatim (please confirm)

Line 113: In the analysis section, was member checking performed to confirm research trustworthiness. In other words, were the participants shown a copy of the transcript to confirm accuracy.

Line 130 Better to use 'Findings' for qualitative research.

Author Response

(The authors gave the same response as above.)
